# Distinct Inflammatory Responses of hiPSC-Derived Endothelial Cells and Cardiomyocytes to Cytokines Involved in Immune Checkpoint Inhibitor-Associated Myocarditis

**DOI:** 10.3390/cells14171397

**Published:** 2025-09-07

**Authors:** Samantha Conte, Isaure Firoaguer, Simon Lledo, Thi Thom Tran, Claire El Yazidi, Stéphanie Simoncini, Zohra Rebaoui, Claire Guiol, Christophe Chevillard, Régis Guieu, Denis Puthier, Franck Thuny, Jennifer Cautela, Nathalie Lalevée

**Affiliations:** 1Aix Marseille Univ, Inserm, INRAE, C2VN, 13005 Marseille, France; samantha.conte@etu.univ-amu.fr (S.C.); isaure.firoaguer@univ-amu.fr (I.F.); simon.lledo@etu.univ-amu.fr (S.L.); thi-thom.nguyen@univ-amu.fr (T.T.T.); stephanie.simoncini@univ-amu.fr (S.S.); zohra.rebaoui@etu.univ-amu.fr (Z.R.); claire.guiol@univ-amu.fr (C.G.); regis.guieu@univ-amu.fr (R.G.); franck.thuny@gmail.com (F.T.); jennifer.cautela@ap-hm.fr (J.C.); 2MarMaRa Institute, 13385 Marseille, France; claire.elyazidi@univ-amu.fr (C.E.Y.); christophe.chevillard@univ-amu.fr (C.C.); denis.puthier@univ-amu.fr (D.P.); 3ICI Institute, 13005 Marseille, France; 4Aix Marseille Univ, Inserm, MMG, 13385 Marseille, France; 5Aix Marseille Univ, Inserm, TAGC, 13005 Marseille, France; 6Turing Centre for Living Systems, 13005 Marseille, France; 7Transcriptomics and Genomics Marseille Luminy (TGML), 13013 Marseille, France; 8Aix-Marseille Univ, AP-HM, University Mediterranean Center of Cardio-Oncology, Unit of Heart Failure and Valvular Heart Diseases, Department of Cardiology, North Hospital, 13915 Marseille, France

**Keywords:** inflammatory cytokines, hiPSC-EC, hiPSC-CM, ICI-induced myocarditis, inflammasome, apoptosis

## Abstract

Inflammatory cytokines, particularly interferon-γ (IFN-γ), are markedly elevated in the peripheral blood of patients with immune checkpoint inhibitor-induced myocarditis (ICI-M). Endomyocardial biopsies from these patients also show GBP-associated inflammasome overexpression. While both factors are implicated in ICI-M pathophysiology, their interplay and cellular targets remain poorly characterized. Our aim was to elucidate how ICI-M-associated cytokines affect the viability and inflammatory responses of endothelial cells (ECs) and cardiomyocytes (CMs) using human induced pluripotent stem cell (hiPSC)-derived models. ECs and CMs were differentiated from the same hiPSC line derived from a healthy donor. Cells were exposed either to IFN-γ alone or to an inflammatory cytokine cocktail (CCL5, GZMB, IL-1β, IL-2, IL-6, IFN-γ, TNF-α). We assessed large-scale transcriptomic changes via microarray and evaluated inflammatory, apoptotic, and cell death pathways at cellular and molecular levels. hiPSC-ECs were highly sensitive to cytokine exposure, displaying significant mortality and marked transcriptomic changes in immunity- and inflammation-related pathways. In contrast, hiPSC-CM showed limited transcriptional changes and reduced susceptibility to cytokine-induced death. In both cell types, cytokine treatment upregulated key components of the inflammasome pathway, including regulators (GBP5, GBP6, P2X7, NLRC5), a core component (AIM2), and the effector GSDMD. Increased GBP5 expression and CASP-1 cleavage mirrored the findings found elsewhere in endomyocardial biopsies from ICI-M patients. This hiPSC-based model reveals a distinct cellular sensitivity to ICI-M-related inflammation, with endothelial cells showing heightened vulnerability. These results reposition endothelial dysfunction, rather than cardiomyocyte injury alone, as a central mechanism in ICI-induced myocarditis. Modulating endothelial inflammasome activation, particularly via AIM2 inhibition, could offer a novel strategy to mitigate cardiac toxicity while preserving antitumor efficacy.

## 1. Introduction

Aberrant cytokine release by activated immune cells, particularly in the context of cancer immunotherapy, is a critical driver of immune-mediated myocarditis. Pro-inflammatory cytokines secreted by leukocytes and other immune effector cells orchestrate inflammatory cascades that underlie both autoimmune and cardiovascular pathologies [1]. The advent of immune checkpoint inhibitors (ICIs) has revolutionized cancer therapy by reactivating the immune system ability to recognize and attack tumors. These monoclonal antibodies inhibit receptor-ligand interactions that suppress T cell activation, thereby enhancing immune responses. However, the disruption of immunological self-tolerance can lead to side effects such as inflammation and immune-related adverse events (irAEs) [2]. Among these, cardiovascular disorders such as vasculitis, pericarditis, arrhythmias, and myocarditis represent severe complications, often associated with high mortality [3,4]. In cases of ICI-induced myocarditis, specific pathogenic immune cell subsets exhibiting distinct transcriptional profiles, including the upregulation of chemokines like CCL5, CCL4, and CCL4L2, have been identified [5]. Elevated circulating levels of TNF-α, IFN-γ, GM-CSF, IL-1α, IL-1β, and IL-6 have also been reported in cancer patients with irAEs [6,7,8]. However, their utility in monitoring ICI-induced myocarditis remains limited [6,9].

The diagnosis of ICI-induced myocarditis currently relies on endomyocardial biopsy and the detection of myocardial immune cell infiltrates. However, the underlying pathophysiological mechanisms remain poorly understood [10,11]. Patients receiving ICI therapy exhibit varying grades of inflammatory infiltrate, which may or may not be accompanied by cardiomyocyte loss [12]. Transcriptomic analyses of myocardial biopsies from patients with ICI-induced myocarditis showed transcriptional changes including the upregulation of IFN-γ-responsive pathways. In particular, guanylate-binding proteins GBP5 and GBP6 were significantly increased at both the RNA and protein levels [13]. IFN-γ is known to promote PD-L1 expression on cardiac cells as a compensatory mechanism to limit excessive CD8^+^ T cell activity [14,15]. These findings point to the potential involvement of the NOD-like receptor family, pyrin domain-containing 3 (NLRP3) inflammasome in the pathogenesis of ICI-induced myocarditis, a mechanism that appears distinct from that observed in viral myocarditis [13].

GBPs play a pivotal role in promoting the assembly of the NLRP3 inflammasome. Inflammasomes are sensory complexes that detect infections or tissue damage and initiate inflammatory responses by activating caspase-1, which in turn triggers the maturation and secretion of interleukins IL-1β and IL-18 [16]. In healthy myocardial tissue, the low expression level of NLRP3 components is insufficient to support inflammasome assembly or activation. The initial step in NLRP3 activation requires transcriptional priming, often mediated by NF-κB signaling in response to stimuli such as pro-inflammatory cytokines [17,18]. Within the heart, activation of the NLRP3 inflammasome in monocytes, fibroblasts, and endothelial cells leads to robust secretion of IL-1β. In contrast, activation within cardiomyocytes primarily results in IL-18 release and induction of pyroptosis, a form of inflammatory cell death driven by caspase-1 activation [19].

Previous studies have highlighted key mechanisms underlying immune toxicity, including the release of pro-inflammatory cytokines by ICI-activated T cells within myocardial tissue [5,6,8,10,12,13,20]. However, the diagnosis of immune-related myocarditis remains particularly challenging, and a deeper understanding of its pathophysiology is essential to improve both diagnostic accuracy and therapeutic strategies. While preclinical models have proven invaluable in addressing cardiovascular side effects of ICI therapy, they poorly address the specific mechanisms driving immune-related myocarditis [21]. Moreover, only a limited number of studies have explored the cellular and molecular basis of ICI-induced myocarditis in humans. This is largely due to practical and ethical limitations: large-scale collection of fresh human cardiac tissue is not feasible, and primary human cardiomyocytes are difficult to access. Additionally, endomyocardial biopsies, although informative, are invasive, carry procedural risks, and suffer from low diagnostic sensitivity, further limiting in-depth mechanistic investigations.

Recent advances in the generation of human induced pluripotent stem cells (hiPSCs) and their differentiation into cardiac cells offer a scalable and human-relevant alternative to traditional models [22,23]. Despite this progress, cellular models specifically tailored to study immune-mediated myocarditis remain scarce, and no hiPSC clones have yet been derived from patients. Moreover, research has predominantly focused on the cardiomyocyte component, often neglecting the multicellular complexity of the myocardium. Recently, Jensen et al. developed and characterized a tissue culture–based model of myocarditis using hiPSC-derived cardiomyocytes and PBMCs from multiple healthy donors. Upon exposure to ICIs, this model enabled the reproduction of key pathological features such as cardiomyocyte necrosis, sarcomere disruption, and arrhythmogenic changes, offering new avenues to investigate disease mechanisms in a controlled and reproducible human-based system [24].

To our knowledge, no cellular model of myocarditis has been developed to simultaneously address both cardiomyocyte and endothelial cell properties. Yet, endothelial cells play a critical role in the inflammatory response and are known to exhibit high apoptotic priming throughout life, making them particularly vulnerable to inflammatory insults [25,26]. As a consequence, endothelial toxicity represents a key feature of anticancer therapies, with specific vascular-damaging effects [26]. In the context of immune-mediated myocarditis, a deeper understanding of the mechanistic alterations occurring within the endocardial and microvascular endothelium is essential, particularly in relation to immune cell infiltration and local cytokine release. Preserving or restoring endothelial integrity could prove crucial in mitigating cardiac damage. In this study, we established a hiPSC-derived model from a healthy donor to generate both cardiomyocytes and endothelial cells from the same clone, thereby minimizing genetic variability and enabling direct comparison of cell-type-specific responses. This model provides a unique framework to investigate the pathophysiological mechanisms triggered by pro-inflammatory cytokines implicated in ICI-induced myocarditis. This work aims to explore the transcriptional and cell death responses induced by inflammatory cytokines, primarily IFN-γ, which are found at elevated levels in both myocardial biopsies and peripheral blood of affected patients. By assessing markers of inflammation, apoptosis, and pyroptosis, we also directly compared the effects of cytokine exposure on endothelial cells and cardiomyocytes derived from the same hiPSC clone enabling the assessment of cell-type-specific responses.

## 2. Materials and Methods

### 2.1. Biological Samples and Cell Culture

The human induced pluripotent stem cell (hiPSC) clone ECT-06 was established by the cell reprogramming and differentiation facility (MaSC) of the Marseille Medical Genetics Institute (MMG-UMR 1251), from skin fibroblasts from a healthy male donor (aged 69, reference AG16102) from the Coriell Institute Cell Repository (Camden, NJ, USA) (CODECOH N° DC-2018-3207 for IPSC, MTA signed in 2016 for the use of fibroblasts). The iPSC clone was derived from primary fibroblasts generated after the transfection of different vectors via electroporation (pCXCLE-hOCT3/4-shp53-F (Addgene ref 27077), pCXLE-hSK (Addgene, ref 27078, encoding *SOX2* and *KLF4*), pCXLE-hUL (Addgene, ref 27080, encoding *L-Myc* and *LIN28*). Human iPS (hiPSC) colonies were picked 4 to 6 weeks after transfection based on their ES cell-like morphology. Colonies were grown and expanded in mTeSR1 medium (StemCells) on Matrigel (Corning, Avon, France) coated dishes. The ECT-06 clone was fully characterized using classical procedures recommended by the ISCCR, such as checking the expression of pluripotency markers and for the absence of the expression of reprogramming transgenes, as well as phosphatase alkaline staining and karyotype to check for the absence of chromosomal abnormalities.

hiPSCs were cultured in mTeSR1 medium (STEMCELL Technologies, Vancouver, BC, Canada) on Matrigel-coated plates. Cells were dissociated every 5–6 days using Versene (ThermoFisher Scientific, Waltham, MA, USA). Prior to differentiation, cells were dissociated with Gentle Cell Dissociation reagent (STEMCELL Technologies) supplemented with RevitaCell (ThermoFisher Scientific). HiPSC-derived cardiomyocytes (hiPSC-CM) were generated using the STEMdiff™ Ventricular Cardiomyocyte Differentiation and Maintenance Kit (STEMCELL Technologies) according to the manufacturer’s instructions. On day 15 of differentiation (D15), cells were dissociated and cultured in Maintenance medium (STEMCELL Technologies) on Matrigel. hiPSC-derived endothelial cells (hiPSC-EC) were generated using the STEMdiff™ Endothelial Differentiation Kit (STEMCELL Technologies). Following differentiation, hiPSC-ECs were dissociated weekly using TrypLE and cultured on 0.01% gelatin in MV2 medium supplemented with 25 ng/mL VEGF. Assays were performed on hiPSC-CM on day 28 of differentiation (D28) and on hiPSC-EC at passage 4. AC16 cells (Merck, Rahway, NJ, USA) were cultured in DMEM/F-12 medium (ThermoFisher Scientific) supplemented with 12.5% FBS. Dissociation was performed using TrypLE (ThermoFisher Scientific). Human cardiac microvascular endothelial cells (HCMECs) were cultured in MV2 medium (PromoCell, Heidelberg, Germany) and dissociated with Accutase (ThermoFisher Scientific).

### 2.2. Cell Treatment and Sample Preparation

Three independent differentiations of hiPSC-derived endothelial cells (hiPSC-EC) and cardiomyocytes (hiPSC-CM) were generated for microarray analysis. Within each differentiation, cells were split into two groups: one exposed to IFN-γ (10 ng/mL; Miltenyi Biotec, Paris, France) for 72 h, and the other left untreated as the control. Each sample corresponded to a pool of three wells of cultured cells.

As 72 h of IFN-γ exposure resulted in high mortality in hiPSC-ECs, the treatment duration was reduced to 48 h for all subsequent experiments, while maintaining the same cytokine concentration, in order to preserve the cell number for downstream assays. For these additional analyses, three new differentiations were performed for each cell type. One hiPSC-CM differentiation did not meet quality control criteria, and the results were excluded. All experimental assays (cell viability, qPCR, immunostaining, and flow cytometry) were conducted on the same batch of cells to allow direct comparison across methods. For cytometry and qPCR, three wells per differentiation were pooled to obtain sufficient material. For viability assays and immunostaining of GBP5 and CASP1, three and five wells per differentiation, respectively, were analyzed. Viability and qPCR results represent the mean of three technical replicates.

For dose–response experiments with single pro-inflammatory molecules, cells were treated for 48 h with CCL5 (10 ng/mL; BioLegend, San Diego, CA, USA), IL-1β (10 ng/mL; BioLegend), IL-2 (2 ng/mL; Miltenyi Biotec), or IL-6 (10 ng/mL; BioLegend). In all other experiments, cells were exposed for 48 h to a cytokine cocktail containing CCL5, IL-1β, IL-2, IL-6, or IFN-γ (same concentrations as above), in addition to GZMB (5 ng/mL; Merck) and TNF-α (5 ng/mL; Miltenyi Biotec). Camptothecin (3 µg/mL; Merck) was used as a positive control for caspase activation.

For each cell type, including all hiPSC-derived cells, a pool of untreated cells was included as a negative control in every experiment.

### 2.3. RNA Extraction

Total RNA was extracted using the RNeasy^®^ Mini Kit (QIAGEN, Hilden, Germany) after homogenization with QIAshredder columns (QIAGEN). RNA was treated with RNase-Free DNase Set (#79256, QIAGEN) following the manufacturer’s instructions. The concentration of RNA was determined by reading the absorbance at 260 nm using a NanoDrop (ND-1000, ThermoScientific, Waltham, MA, USA). The ratio of A260/280 in all the RNA samples ranged from 1.8 to 2.0. The quality was confirmed using an Agilent 2100 Bioanalyzer (Agilent Technologies, Santa Clara, CA, USA) with Agilent RNA 6000 Nano Chips. Samples with an RIN higher than 8 were used in microarray and qPCR experiments.

### 2.4. Microarray

Gene expression was measured using a SurePrint G3 Human Gene Expression v3 8 × 60K Microarray Kit (Agilent Technologies) containing 629,976 oligonucleotide probes representing 44,782 genes. Total RNAs (200 ng) were labeled with cyanine 3-CTP using a Low RNA Input Quick Amp Labelling, One color Kit according to the manufacturer’s protocol (Agilent Technologies, CA, USA). For each reaction, the cRNA yield and specific activity of cRNA were determined using a NanoDrop ND-1000 spectrophotometer. Only cRNAs with yields > 0.9 μg and specific activities > 6.0 pmol of dye per microgram of cRNA were used for hybridization. The labeled cRNAs were hybridized to microarray slides (eight arrays per slide) following the Agilent One-Color microarray-based gene expression analysis protocol (Agilent Technologies). The slides were scanned (8 × 60K array slides at 3 μm resolution) using an Agilent DNA microarray scanner (G2505C) and with the color settings set to those of the Agilent G3_GX_1 protocol. The scanned images were analyzed using Feature Extraction Software 10.5 (Agilent). In total, 12 raw data files (3 arrays for hiPSC-CM CTRL and 3 arrays for hiPSC-CM IFN, 3 arrays for hiPSC-EC CTRL, and 3 arrays for hiPSC-EC IFN) were obtained using the Agilent Feature Extraction Software. The raw intensities were exported in Excel format and converted in logarithmic scale (base 2). Quantile normalization was applied using RStudio (1.2.5033-1) to correct for global intensity and dispersion. Then, stringent filtering was applied to retain only the probes that showed a signal above background in all three technical replicates of at least one experimental condition (CTRL or IFN), for any cell type. To detect significant variation in gene expression between groups (hiPSC-CM CTRL vs. IFN and hiPSC-EC CTRL vs. IFN), the significance analysis of microarrays (SAM) method was used, with 10,000 permutations applied and a false discovery rate (FDR) set to 0%. The hierarchical clustering of differentially expressed genes (DEGs) was achieved using the TM4 Microarray Software Suite V4.9 (http://mev.tm4.org (accessed on 13 March 2020)). Distance was computed based on Pearson’s correlation coefficient, and agglomerative criteria was set to “average”. For a biological interpretation of the gene expression data, functional enrichment analyses of up- and down-regulated genes were performed using g:profiler [27]. Biological processes, cellular components and Kyoto Encyclopedia of Genes and Genomes (KEGG) pathways were considered significantly enriched when the adjusted *p*-value (Benjamini–Hochberg) was below 0.05.

### 2.5. Cell Viability Assays

Cell viability was assessed using the Deep Blue Cell Viability™ kit (BioLegend) according to the manufacturer’s instructions. Fluorescence was measured using a GloMax^®^ Discover system (Promega, Madison, WI, USA) (excitation: 520 nm; emission: 580–640 nm).

### 2.6. Real-Time Quantitative PCR (qPCR)

Reverse transcription was performed using the SuperScript^®^ VILO™ Master Mix kit (ThermoFisher Scientific). qPCR reactions were performed using the Power SYBR Green Master Mix^®^ (Thermo Fisher Scientific) on a StepOne Plus Real-Time qPCR system (Applied Biosystems, Foster City, CA, USA). Data were analyzed with StepOne software v2.3 (Applied Biosystems). All primers used had an efficiency between 95 and 105% (Table A1). Expression levels were normalized using the geometric mean of reference genes RPL13A and YWHAZ. −∆CT values were calculated for statistical analysis.

### 2.7. Flow Cytometry

hiPSC-derived CM and EC differentiation was characterized by quantifying specific proteins: Troponin T2 and myosin heavy chain for hiPSC-CMs, and CD31 and VE-Cadherin for hiPSC-ECs. hiPSC-CMs were fixed and permeabilized using the eBioscience™ Transcription Factor Staining Buffer Set (ThermoFisher Scientific). Samples with >80% positive cells for specific markers were used in subsequent studies. The expression of MHC-I (Miltenyi Biotec), MHC-II (Miltenyi Biotec), and PD-L1 (Miltenyi Biotec) was quantified. Cell viability was monitored using ViaKrome 808 Fixable Viability Dye (Beckman Coulter, Brea, CA, USA) for hiPSC-CM, and Viobility 405/452 Fixable Dye (Miltenyi Biotec) for hiPSC-EC, AC16, HCAEC, and HCMEC. Caspase-3/-7 activity was measured using the CellEvent™ Caspase-3/7 kit (ThermoFisher Scientific) in living and dying cells identified using the LIVE/DEAD™ Fixable Dead Cell Stain Kit (ThermoFisher Scientific).

### 2.8. Immunofluorescence Staining

hiPSC-EC and hiPSC-CM were seeded on Lab-Tek 8 chambers coated using 0.1% gelatin and Matrigel, respectively. Cells were fixed using 4% paraformaldehyde (PFA), permeabilized using 0.2% Triton X-100 (Sigma-Aldrich, St. Louis, MO, USA), and blocked using bovine serum albumin (Merck). hiPSC-ECs were characterized using primary antibodies against CD31 (polyclonal rabbit IgG, 1:200, Proteintech, Rosemont, IL, USA) and vWF (monoclonal mouse IgG, 1:100, Proteintech). hiPSC-CMs were characterized using primary antibodies against TNNT2 (monoclonal mouse IgG1, 1:200, ThermoFisher Scientific) and Ki67 (polyclonal rabbit IgG, 1:500, PA5-16785, ThermoFisher Scientific). hiPSC-ECs and hiPSC-CMs treated using IFN-*γ* or cytokine cocktail, we addressed two inflammasome-related proteins: Caspase-1/p20/p10 (rabbit polyclonal IgG, 1:100, Proteintech) and GBP5 (rabbit polyclonal IgG, 1:200, Proteintech). Secondary antibodies used were Alexa Fluor 594 goat anti-mouse IgG (1:1000, Thermo Fisher Scientific) and Alexa Fluor 488 goat anti-rabbit IgG (1:1000, Thermo Fisher Scientific). Images were acquired using a Zeiss Axio Observer microscope and analyzed using ZEN Blue software (v3.7.97.03000, Carl Zeiss, Oberkochen, Germany). Fluorescence intensity was quantified using ZEN software. Normalized fluorescence values for GBP5 and CASP1 were obtained by measuring the total fluorescence signal in each well and dividing it by the corresponding total cell number. Cytoplasmic GBP5 expression was assessed by counting the cells manually, with all cells exhibiting detectable cytosolic GBP5 staining, irrespective of signal intensity. The number of positive cells was then normalized vs. the total cell count and reported as a percentage.

### 2.9. Statistical Analysis

Data are expressed as mean ± standard deviation (SD). We used the Shapiro–Wilk test to verify the normal distribution of the samples, then significant differences between groups were determined using an unpaired *t*-test, with Welch’s correction applied when appropriate, using GraphPad Prism software (GraphPad Prism 9.0, La Jolla, CA, USA). A *p* < 0.05 was considered statistically significant.

## 3. Results

### 3.1. Characterization of hiPSC-Derived ECs and CMs

To investigate how inflammatory molecules from the immune infiltrate affect cardiac cells, we used the same hiPSC clone from a healthy donor to generate both endothelial cells and cardiomyocytes, minimizing potential biases associated with donor variability or the use of primary cardiac cell lines and cultures. We first characterized the subpopulations obtained after differentiation. In the hiPSC-derived endothelial cell (hiPSC-EC) cultures, most cells (90 ± 5%) expressed CD31 and VE-cadherin. All of these also expressed endoglin (CD105) and CD54, a capillary-specific marker of vascular ECs (Figure 1A). The cells displayed a typical spindle shape and network organization. Immunostaining confirmed CD31 localization at the plasma membrane and cytosolic expression of vWF, a vascular endothelial marker (Figure 1B). The endocardium, which interfaces with both the bloodstream and sub-endocardial conducting cells, plays a critical role in immune cell infiltration during immune-mediated myocarditis, leading to conduction disorders. Among the hiPSC-EC subpopulations, approximately 60% expressed endocardial markers such as CD44 (aortic valve component) and integrin α9 (ITGA9) (Figure 1A). In hiPSC-derived cardiomyocyte (hiPSC-CM) cultures, spontaneous beating was observed 8–10 days post-differentiation (33 ± 7 bpm, N = 9). The majority (85 ± 3%) expressed TNNT2, and most TNNT2^+^ cells (81 ± 4%) also expressed myosin heavy chain (MYH), indicating a high degree of maturation. Among MYH^+^ cells, 53 ± 15% expressed atrial myosin light chain, 23 ± 24% expressed the ventricular form, and 15 ± 17% co-expressed both (Figure 1C). Morphologically, the cells showed syncytial organization and prominent sarcomeres. Only a few Ki67^+^ cells remained undifferentiated (Figure 1D).

### 3.2. Transcriptomic Effect of IFN-γ in hiPSC-Derived Cardiac Cells

IFN-γ is a cytokine released during inflammation that activates multiple signaling pathways, including Toll-like receptor (TLR) and MHC-associated immune pathways. In myocarditis, IFN-γ induces PD-L1 overexpression. To mimic this inflammatory environment, we treated hiPSC-ECs and hiPSC-CMs with IFN-γ (10 ng/mL, 72 h).

We analyzed transcriptomic changes post-treatment. In hiPSC-ECs, we identified 1201 differentially expressed genes (DEGs): 863 upregulated and 338 downregulated (Figure 2A,B; Appendix A—Gene lists). In contrast, hiPSC-CMs showed only 91 DEGs, with 90 upregulated and 1 downregulated gene, and none associated with significant biological processes (Figure 2A,B and Appendix A—Gene lists). Both cell types shared 80 upregulated DEGs related to immune processes (Figure 2A,B and Appendix A—FunctAnnot-Common Up-reg genes). In hiPSC-ECs, enriched biological processes included immune responses, IFN-γ signaling, inflammasome activation, cell death, metabolism, and inflammatory signaling (Figure 2C). Cell death, apoptosis, proliferation, development, and migration processes were also specific to ECs (Figure 2C). Downregulated genes in ECs were mainly linked to cell adhesion and communication (Appendix A—FunctAnnot-EC Down-reg genes).

We then focused on three main biological processes (Figure 3): immunity/inflammation, cell death, and adhesion. Most transcriptional changes occurred in hiPSC-ECs. For immunity/inflammation, *Programmed Cell Death 1 Ligand 1 and 2* (*PD-L1/PD-L2*) and *Interleukin 1 Receptor* (*IL-1R*) were upregulated. *IL-1R* is associated with the *Mitogen-Activated Protein Kinase* (*MAPK pathway*). We also observed overexpression of *Major Histocompatibility Complex I and II* (*MHC-I/II*) and *Toll-Like Receptor 4* (*TLR4*), linked to *Nuclear Factor-kappa B* (*NF-κB*) signaling, and upregulation of the *Janus Kinase Signal Transducer and Activator of Transcription* (*JAK/STAT*) pathway. These pathways converge to activate *Interferon Regulatory Factors* (*IRF*)*1*, *7*, *8*, *and 9*, which regulate vascular permeability and promote immune cell infiltration [28].

Infiltration can also be enhanced through the overexpression of *Intercellular Adhesion Molecule 1* (*ICAM-1)* and *Carcinoembryonic Antigen-Related Cell Adhesion Molecule 1* (*CEACAM1)*, which facilitate leukocyte adhesion and transmigration, as well as *Collagen type XIII alpha 1 chain* (COL13A1), potentially contributing to inflammation. These transcriptional changes are indicative of disrupted endothelial function [28] and were associated with increased production of inflammatory cytokines: *C-X-C motif Chemokine ligand 9, 11, and 16* (*CXCL9/11/16*), *C-X3-C motif Chemokine ligand 1* (*CX3CL1*), *C-C motif Chemokine ligand 2 and 8* (*CCL2/8*), *Interleukine alpha* (*IL-1A*), and *Granzyme A* (GZMA).

In hiPSC-CMs, transcriptional changes were modest. However, *MHC-I* and *STAT1* were upregulated, activating *IRF8* and *IRF9*, and inducing *CXCL9* expression, suggesting leukocyte migration into the myocardium. *CEACAM1* was also upregulated, though its cardiac role remains unclear [29].

In the analysis of cell death and inflammatory signaling pathways, hiPSC-ECs demonstrated a distinct upregulation of *Absent in Melanoma 2* (*AIM2*), a key cytosolic DNA sensor and inflammasome activator. This *AIM2* overexpression, not observed in hiPSC-CMs, highlights a potential endothelial-specific immune response to cytoplasmic DNA stress. In addition, hiPSC-ECs also showed exclusive increases in *Fibroblast Activated Sequence/Caspase 10* (FAS/CASP10), GBP4, P2RX7, and TXNIP, implicating both apoptotic and inflammasome pathways. In both cell types, several NLRP3-related regulators were upregulated, including *Guanylate-binding proteins* (*GBP*) *1*, *GBP2*, *GBP3*, *GBP5*, and *NLR Family CARD Domain Containing 5* (*NLRC5)*, with the latter acting as a shared modulator of NLRP3/AIM2 activity and NF-κB signaling.

This transcriptional priming led to *Caspase 1* (*CASP-1*) overexpression in both cell types. However, the expression of downstream inflammasome effectors, *Gasdermin* (*GSDM) B/D* and *Interleukin 1 beta* (*IL-1β*), was restricted to hiPSC-ECs (Figure 3). CASP-1 cleaves GSDMD, whose N-terminal domain forms membrane pores, triggering pyroptosis. IL-1β release promotes acute inflammation [30]. These results show that IFN-γ treatment induces the activation of inflammation and cell death in both cell types although at the transcriptional level, activation occurs predominantly in hiPSC-ECs.

### 3.3. Effect of IFN-γ and Cytokine Cocktail on Cell Mortality and Inflammatory Pathways

To mimic the inflammatory environment found in myocardium from patients with post-ICI immune-mediated myocarditis, we exposed hiPSC-ECs and hiPSC-CMs to a cocktail of inflammatory cytokines identified in patient plasma/serum and biopsies [5,6,8,9,13]. We first optimized the concentration of each cytokine by evaluating their effects on cell viability and expression of PD-L1 and MHC-I in hiPSC-ECs, hiPSC-CMs, and AC16 ventricular cardiomyocytes (Figure A1 and Figure A2). For each cytokine, we selected the concentration that enhanced PD-L1 and MHC-I expression while maintaining moderate mortality (20–30%). The final cytokine cocktail included IL-1β, IL-6, CCL5, and IFN-γ (10 ng/mL each), IL-2 (2 ng/mL), and TNF-α and GZMB (5 ng/mL each).

We then compared cellular responses to IFN-γ alone versus the cytokine cocktail. Both treatments induced a trend toward increased PD-L1 and MHC-I expression in hiPSC-ECs and hiPSC-CMs. However, due to the variability in responses, differences were not statistically significant (Figure 4A,B). We next assessed cell mortality. In hiPSC-ECs, IFN-γ alone caused substantial cell death (45 ± 9%), whereas hiPSC-CMs remained unaffected. The cytokine cocktail had no effect on hiPSC-CM viability but significantly increased hiPSC-EC mortality (89 ± 7%), nearly doubling that observed with IFN-γ alone (Figure 4C,D). These experiments were also conducted using AC16 cardiomyocytes and primary cardiac microvascular endothelial cells (HCMEC), confirming similar responses as seen with hiPSC-derived cells (Figure A3A,B).

### 3.4. Effect of IFN-γ and Cytokine Cocktail on NLRP3 Inflammasome Regulation

We first examined IFN-γ-induced upregulation of NLRP3 inflammasome-associated transcripts using qRT-PCR, followed by testing with the cytokine cocktail. In hiPSC-ECs, *GBP5* and *GBP6* expression increased with IFN-γ; *GBP5* was further upregulated by the cocktail (Figure 5). In hiPSC-CMs, IFN-γ induced a trend toward increased *GBP5/GBP6*, which seemed to be amplified with *GBP5* after the cytokine cocktail treatment (Figure 5).

At the protein level, immunostaining revealed GBP5-positive perinuclear vesicles under control conditions in both cell types. After IFN-γ treatment, 40 ± 16% of hiPSC-ECs and 7 ± 5% of hiPSC-CMs exhibited strong cytosolic GBP5 expression. The cytokine cocktail further increased this effect (73 ± 46% in hiPSC-EC, 17 ± 11% in hiPSC-CM), coinciding with elevated mortality in hiPSC-EC (Figure 6).

*P2X7* mRNA levels increased in hiPSC-EC with IFN-γ, but the cytokine cocktail had no further effect. In hiPSC-CMs, IFN-γ increased *P2X7* expression, while the cocktail reduced it (Figure 5). Similar patterns were observed in AC16 cardiomyocytes. In contrast, primary HCMEC endothelial cells consistently upregulated *P2X7* in both conditions (Figure A4). Transcriptomic analysis did not detect *P2X7* changes in hiPSC-CM post-IFN-γ (Figure 2), likely due to stringent FDR criteria excluding variable replicates. NLRC5 was upregulated by IFN-γ in hiPSC-ECs, a similar trend was seen in hiPSC-CMs, with no additional effect from the cytokine cocktail (Figure 5). *NLRP3* expression remained unchanged regardless of treatment or cell type (Figure 5), consistent with transcriptomic data (Figure 3). *AIM2* was upregulated by IFN-γ in hiPSC-ECs and slightly more so with the cytokine cocktail; a similar trend was seen in hiPSC-CMs (Figure 5). *GSDMD* expression increased in hiPSC-EC after IFN-γ, a similar trend was seen in hiPSC-CM, with no further change from cytokine cocktail treatment (Figure 5).

Together, these results support the data obtained from the transcriptomic analysis, indicating that most inflammasome-related changes are driven by IFN-γ. Only AIM2 and GBP5 were further enhanced by the cytokine cocktail in both cell types. P2X7 displayed a differential response in hiPSC-CMs, with IFN-γ inducing its expression and the cocktail suppressing it.

### 3.5. Effect of IFN-γ and Cytokine Cocktail on Caspase Activity

To identify mechanisms of cytokine-induced cell death, particularly in hiPSC-ECs, we assessed caspase activation. CASP-1, activated by the NLRP3 inflammasome, drives pyroptosis. Immunostaining showed IFN-γ strongly increased cleaved CASP-1 levels in both hiPSC-ECs and hiPSC-CMs (139% and 61%, respectively), with further elevation in hiPSC-ECs after cytokine cocktail exposure (467%) (Figure 7). Importantly, expression was substantially higher in hiPSC-ECs than in hiPSC-CMs (Figure 7B).

Apoptosis, a non-inflammatory form of programmed cell death, involves CASP-3/-7. We measured their activity separately in living (membrane intact) and dying (membrane permeable) cells. In hiPSC-CMs, basal CASP-3/-7 activity was low (<10%), and IFN-γ did not affect it. However, the cytokine cocktail treatment significantly increased activity, reaching 20%, including 12% dying cells, consistent with the modest mortality observed (Figure 4D and Figure 8).

In hiPSC-ECs, CASP-3/-7 activity remained around 20–22% across conditions. However, while IFN-γ had little effect, the cytokine cocktail induced a strong rise in dying cells showing CASP-3/-7 activation, reaching levels comparable to those of the camptothecin treatment, a potent apoptosis inducer via DNA topoisomerase I inhibition (Figure 8). After cytokine cocktail exposure, 46 ± 20% of hiPSC-ECs showed CASP-3/-7 activity, with 89 ± 7% undergoing cell death.

These findings suggest that cytokine-induced death in endothelial cells involves both apoptotic (caspase-dependent) and non-apoptotic (caspase-independent) mechanisms. In contrast, cardiomyocyte death remains limited and is primarily apoptotic when it occurs.

## 4. Discussion

In this study, we investigate the responses of cardiomyocytes and endothelial cells to pro-inflammatory cytokines implicated in immune-mediated myocarditis [5,6,8,20,31,32]. To enable an unbiased comparison, both cell types were derived from the same hiPSC clone, eliminating confounders related to genetic background or cell origin and attributing observed differences solely to cell-type-specific cytokine responses. A key advantage of using hiPSC-derived cardiac cells lies in their capacity to reflect cardiac phenotypic diversity. The hiPSC-CM population includes atrial- and ventricular-like subtypes, allowing the assessment of chamber-specific responses. Similarly, hiPSC-ECs displayed distinct vascular identities. We identified subpopulations expressing markers, such as CD105 and CD54, enriched in blood capillaries; ITGA9, specific to endocardial cells; and CD44, notably expressed in the aortic valve [33,34,35]. This model, therefore, provides a pathophysiologically relevant platform that partially recapitulates the cellular and tissue-specific features of the human myocardium, supporting investigation into cell-type-specific responses during immune-mediated myocardial inflammation.

To model the inflammatory environment of immune-mediated myocarditis, we designed a cytokine cocktail composed of pro-inflammatory mediators commonly implicated in this condition: IL-1β, IL-2, IL-6, IFN-γ, TNF-α, and granzyme B [6,7,8]. Among these, IFN-γ emerged as a central cytokine in the context of immune-mediated cardiac inflammation. Finke et al. showed significant upregulation of the IFN-γ response pathway in endomyocardial biopsies from patients with immune-mediated myocarditis. In particular, GBP5 and GBP6 were elevated in these samples compared to viral myocarditis, and absent in biopsies from patients with dilated cardiomyopathy [13]. The pathogenic role of IFN-γ has been reinforced by multiple studies, in both patient cohorts and preclinical models of immune checkpoint inhibitor (ICI)-induced myocarditis, highlighting its role in cardiac immune toxicity [3,10,14,31,36]. Given its central role, we conducted a focused transcriptomic analysis of hiPSC-derived cardiomyocytes and endothelial cells treated with IFN-γ alone and compared the responses to those elicited using a cytokine cocktail. Strikingly, the gene expression profiles indicated that IFN-γ was the predominant driver of the transcriptional changes observed.

Our findings demonstrate that endothelial cells are markedly more sensitive than cardiomyocytes to inflammatory cytokines. hiPSC-ECs underwent extensive transcriptomic remodeling, especially in response to IFN-γ, whereas hiPSC-CMs showed fewer gene expression changes, lower apoptosis rates, and reduced cell death.

In hiPSC-ECs, IFN-γ activated key immune and inflammatory pathways, including MAPK, NF-κB, and JAK/STAT cascades, which converge on interferon regulatory factors (IRFs). These regulators influence vascular permeability by modulating inflammatory mediators and endothelial adhesion molecules [37]. Accordingly, we observed upregulation of ICAM-1 and CEACAM1, two key mediators of leukocyte adhesion and transendothelial migration. This pro-inflammatory shift was accompanied by disrupted intercellular communication and increased production of diverse cytokines and chemokines, consistent with an activated endothelial phenotype. Importantly, several of these transcriptional changes mirror those identified in single-cell RNA sequencing (scRNA-seq) datasets from cardiac biopsies of patients with ICI-M [20]. Specifically, CXCL9, CXCL11, GBP4/5, and STAT1 were enriched in capillary and inflamed endothelial subsets, while STAT1, CX3CL1, and ICAM1 were also upregulated in cardiomyocytes. In ICI-treated Ldlr^−/−^ mice, endothelial VCAM-1 and ICAM-1 were similarly upregulated in the arterial endothelium, potentially facilitating T cell recruitment and accelerating atherosclerotic plaque progression [38]. In another murine model, LAG3 inhibition led to increased IFN-γ production by T cells within plaques, suggesting that checkpoint blockade amplifies local inflammation and promotes plaque destabilization [39]. These results support the translational relevance of our model, indicating its capacity to mimic critical aspects of immune-mediated endothelial and cardiac dysfunction observed in clinical settings.

Apoptosis is a regulated, non-inflammatory form of cell death mediated primarily by caspase-3 and -7 (CASP-3/-7), which induce cell death through the proteolysis of structural and regulatory proteins. In our study, while CASP-3/-7 transcript levels remained unchanged in hiPSC-EC, IFN-γ treatment induced marked upregulation of upstream apoptotic regulators FAS and CASP-10, suggesting the activation of the extrinsic death receptor pathway (N = 3 independent differentiations). Despite no increase in CASP-3/-7 activity under IFN-γ alone, cell mortality reached nearly 50%, indicating that caspase activity is not the cause of cell death (*p* < 0.01, n = 9 from N = 3 independent differentiations). Conversely, the cytokine cocktail treatment strongly elevated CASP-3/-7 activity, associated with a much higher mortality rate approaching 90% (*p* < 0.01 vs. CTRL and vs. IFN-γ, n = 9 from N = 3 independent differentiations). However, elevated CASP-3/-7 activity was also observed under control conditions, possibly due to mechanical stress (e.g., cell detachment) during sample preparation, which may have activated apoptotic pathways. These results suggest that clinically relevant concentrations of inflammatory cytokines, especially IFN-γ, can inflict significant endothelial injury in myocardial tissues, including the endocardium and microvasculature. This disruption likely contributes to the creation of a pro-inflammatory environment surrounding cardiomyocytes, thereby promoting their functional impairment. In contrast, hiPSC-CMs appeared more resilient. IFN-γ induced only modest inflammatory transcriptional changes (N = 3), without significant increases in CASP-3/-7 activity or cell death (n = 6 from N = 2 independent differentiations). Even under cytokine cocktail exposure, although CASP-3/-7 activity rose (*p* < 0.05, N = 2), cell mortality remained low (ns, n = 6 from N = 2 independent differentiations). These observations align with the findings of Jensen et al., who showed that hiPSC-CM cultured in cytokine-rich medium from activated PBMCs (stimulated with anti-CD3/CD28 and ICIs) maintained normal beating frequency and rhythm, without functional impairment [24]. This differential sensitivity between cell types may reflect intrinsic developmental programs. Somatic cells like cardiomyocytes are primed for apoptosis early in life but acquire resistance in adulthood, limiting caspase-mediated death under stress [40]. In contrast, endothelial cells remain sensitive to apoptosis throughout life, making them particularly vulnerable to inflammatory or cytotoxic insults, including those associated with anticancer therapies [26].

The NLRP3 inflammasome and IL-1 cytokines are central to the development and progression of cardiovascular diseases, including inflammatory heart conditions. NLRP3 activation has been implicated in myocardial injury, adverse remodeling, and heart failure [41]. Regardless of a viral, autoimmune, or chemotherapy-induced trigger, many inflammatory cardiac diseases converge on NLRP3 activation and the release of pro-inflammatory cytokines such as IL-1β and IL-18. Clinically, NLRP3 activity correlates with disease severity, particularly in cardiac transplant rejection, where its expression rises with the grade of rejection [42]. These findings highlight the potential of the NLRP3-IL-1 axis as both a biomarker and a therapeutic target. Cardiac sarcoidosis illustrates this well. Characterized by non-caseating granulomas infiltrating the heart, it can lead to conduction abnormalities, arrhythmias, and heart failure. Though its cause is unclear, it likely involves an exaggerated immune response in genetically predisposed individuals. Standard treatments rely on glucocorticoids and immunosuppressants, but their long-term efficacy in cardiac involvement is uncertain. Importantly, NLRP3 activation has been detected in myocardial granulomas, suggesting a key role in pathogenesis and supporting IL-1 blockade as a potential targeted therapy for inflammation-driven myocardial damage [42,43].

Given the pivotal role of IFN-γ in inflammasome regulation, we performed transcriptomic analyses in hiPSC-CM and hiPSC-EC. Inflammasome-related transcripts were broadly upregulated in both cell types, with a more pronounced response in hiPSC-EC. Key regulators (*GBP4*, *TXNIP*) and effectors (*GSDMB*, *GSDMD*, *IL-1β*) were induced, highlighting a robust pro-inflammatory response (N = 3). The purinergic receptor P2X7, a known inflammasome activator, showed cell-type-specific modulation: in hiPSC-CMs, IFN-γ-induced expression was attenuated by the cytokine cocktail (N = 2), suggesting pathway cross-regulation, whereas this modulation was absent in hiPSC-ECs (N = 3). Previous studies have described the context-dependent regulation of P2X7 using cytokines [44,45].

Across conditions, IFN-γ was the primary driver of inflammasome gene activation. Importantly, *NLRP3* transcript levels remained unchanged, pointing to alternative inflammasome activation (N = 3 independent differentiations for both cell types). In contrast, *AIM2* emerged as a central mediator. In hiPSC-ECs, IFN-γ (10 ng/mL) robustly induced *AIM2* expression after 72 h in microarray analysis (N = 3), with qPCR showing significant upregulation already at 48 h (*p* < 0.01, N = 3). This effect was further potentiated by the cytokine cocktail (*p* < 0.001 vs. CTRL; *p* < 0.05 vs. IFN-γ, N = 3). In contrast, IFN-γ alone did not increase *AIM2* expression in hiPSC-CMs (N = 3 for microarray and N = 2 for qPCR), whereas the cytokine cocktail, including IFN-γ, seemed to significantly induce *AIM2* (*p* < 0.05 vs. CTRL, N = 2). Given its role in sensing cytosolic dsDNA and activating IL-1β, IL-18, and GSDMD, AIM2 likely drives endothelial pyroptosis. This is consistent with reports linking AIM2 to endothelial dysfunction in coronary artery disease [46,47] and inflammation-mediated bone loss following chemotherapy [48]. These findings underscore AIM2 as a key contributor to endothelial inflammatory responses and a potential therapeutic target in inflammation-driven cardiovascular disease.

At the protein level, we observed strong upregulation of GBP5 in both hiPSC-ECs (*p* < 0.0001, n = 15 from N = 3 independent differentiations) and hiPSC-CMs following IFN-γ treatment (*p* < 0.01, n = 10 from N = 2 independent differentiations), further amplified by the cytokine cocktail (*p* < 0.01 vs. IFN-γ, n = 15 from N = 3 independent differentiations in hiPSC-ECs; *p* < 0.05 vs. IFN-γ, n = 10 from N = 2 independent differentiations in hiPSC-CMs). GBP5 has been reported to be overexpressed in either cardiomyocytes or capillary endothelial cells of patients with immune-mediated myocarditis, depending on the study [13,20]. Despite pronounced GBP5 induction in hiPSC-CMs, neither IFN-γ nor the cytokine cocktail led to significant cell death (n = 6 from N = 2 independent differentiations). This aligns with prior observations in murine cardiomyocytes, where NLRP3 activation via CaMKIIδ and NF-κB signaling, induced by angiotensin II, did not cause cytotoxicity. CaMKII-driven transcription of inflammatory genes and inflammasome priming occurred within three hours of exposure, preceding macrophage infiltration [49]. These findings suggest that GBP5 upregulation and early inflammasome activation reflect a priming phase rather than direct toxicity. We assessed CASP-1 activity in both cell types, as it mediates pyroptosis downstream of NLRP3. In hiPSC-CMs, IFN-γ and the cytokine cocktail increased cleavage of CASP-1 compared to controls (*p* < 0.01 and *p* < 0.001, n = 10 from N = 2 independent differentiations, respectively). In hiPSC-ECs, IFN-γ alone induced CASP-1 activation (*p* < 0.01 vs. CTRL, n = 15 from N = 3 independent differentiations), further enhanced by the cytokine cocktail (*p* < 0.001 vs. IFN-γ, n = 15 from N = 3 independent differentiations). This indicates a more pronounced inflammasome response in endothelial cells, consistent with their greater sensitivity to inflammatory stress. These results align with a rat model of myosin peptide-induced myocarditis, where immune cell infiltration and NLRP3 upregulation in cardiac tissue were associated with structural remodeling and electrical alterations in the right ventricular outflow tract. This led to increased arrhythmogenicity via calcium dysregulation and upregulation of inflammatory and Ca^2 +^-handling proteins, including NLRP3 [50].

Similarly, Jensen et al. showed that CD8^+^ T cells stimulated with αCD3/CD28 impaired hiPSC-CM function, particularly beat period regularity, an effect worsened by ICIs. Interestingly, dexamethasone preserved cardiomyocyte function and viability, whereas IFN-γ neutralization had no protective effect [24].

Altogether, these data support a central role for endothelial injury in the early stages of myocarditis and cytokine-induced cardiotoxicity. Given the pivotal role of the vasculature in maintaining tissue homeostasis, even subtle endothelial dysfunction can profoundly impair organ function.

### Study Limitations

Differentiation of hiPSCs into mature cardiac cell types is technically demanding and variable. To ensure consistency, each differentiation into endothelial cells and cardiomyocytes was rigorously validated via flow cytometry. This quality control step, though essential, is particularly cell-consuming for non-proliferative cardiomyocytes. Only cultures meeting predefined phenotypic criteria (see Part 1 of the Results) were included, limiting cell availability throughout the study. To maximize use of the material, we first assessed individual cytokines for effects on viability and expression of PD-L1 and MHC-I, before focusing on a cytokine cocktail compared to IFN-γ alone, which emerged as the dominant modulator.

## 5. Conclusions

This study investigated the transcriptional impact of inflammatory cytokines, mainly IFN-γ, on hiPSC-derived cardiac cells in the context of ICI-induced myocarditis. Our findings reveal a marked sensitivity of hiPSC-ECs compared to hiPSC-CMs, characterized by higher mortality and robust transcriptomic remodeling in immune and inflammatory pathways, including MAPK, NF-κB, and JAK/STAT. Cytokine treatment also activated the inflammasome in both cell types. Consistent with observations in endomyocardial biopsies from patients with ICI-induced myocarditis, we observed increased GBP5 protein expression and CASP-1 cleavage following cytokine treatment.

This cellular model offers valuable insight into the early mechanisms of endothelial dysfunction in inflammatory cardiac conditions and its impact on cardiomyocytes. Importantly, data from this healthy donor hiPSC line may serve as a control reference for future studies involving patient-derived hiPSC clones with immune-mediated myocarditis. Such comparative analyses could help identify disease-specific signatures and uncover novel therapeutic targets.

Notably, Burridge et al. previously demonstrated that hiPSC-derived cardiomyocytes from patients with doxorubicin-induced cardiotoxicity recapitulated disease phenotypes and showed transcriptomic changes, including the upregulation of stress-related genes [51]. Similarly, our approach supports the use of patient-specific hiPSC models to dissect the molecular basis of inflammation-driven cardiotoxicity.

## Figures and Tables

**Figure 1 cells-14-01397-f001:**
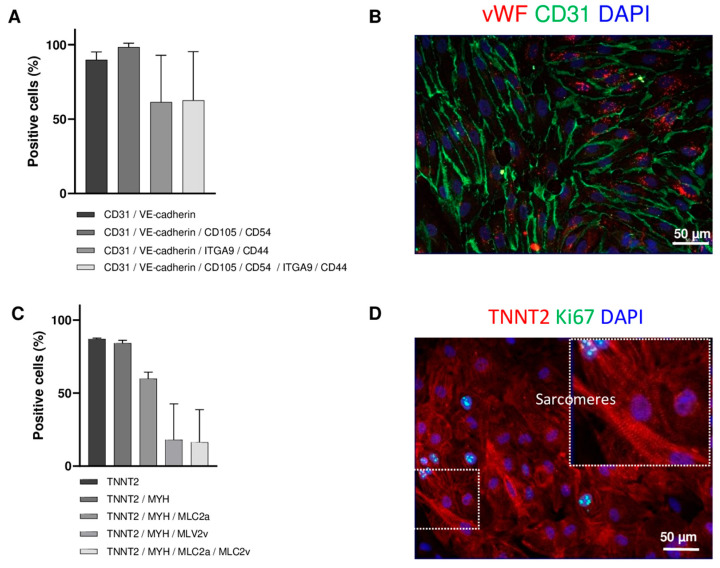
Characterization of hiPSC-derived cardiac cells. (**A**) Co-expression of endothelial markers in hiPSC-derived cells. The percentage of cells co-expressing CD31, VE-cadherin, CD105 (endoglin), CD54, CD44 and ITGA9 (integrin-α9) is shown as mean ± SD (N = 3). (**B**) Immunofluorescence staining of hiPSC-ECs. Expression of the endothelial markers von Willebrand Factor (vWF, red) and CD31 (green) with nuclear staining (DAPI, blue) were studied. (**C**) Co-expression of cardiomyocyte markers in hiPSC-derived cells. The percentage of cells expressing TNNT2 (Troponin T2), MYH (Myosin Heavy Chain), MLC2a and MLC2v (Myosine Light Chain 2 atrial and ventricular), is shown as mean ± SD (N = 2). (**D**) Immunofluorescence staining of hiPSC-CMs. Expression of the cardiomyocyte marker TNNT2 (red) was assessed alongside the proliferation marker Ki67 (green), with DAPI used for nuclear staining (blue). TNNT2 staining reveals the presence of sarcomeres in hiPSC-CMs.

**Figure 2 cells-14-01397-f002:**
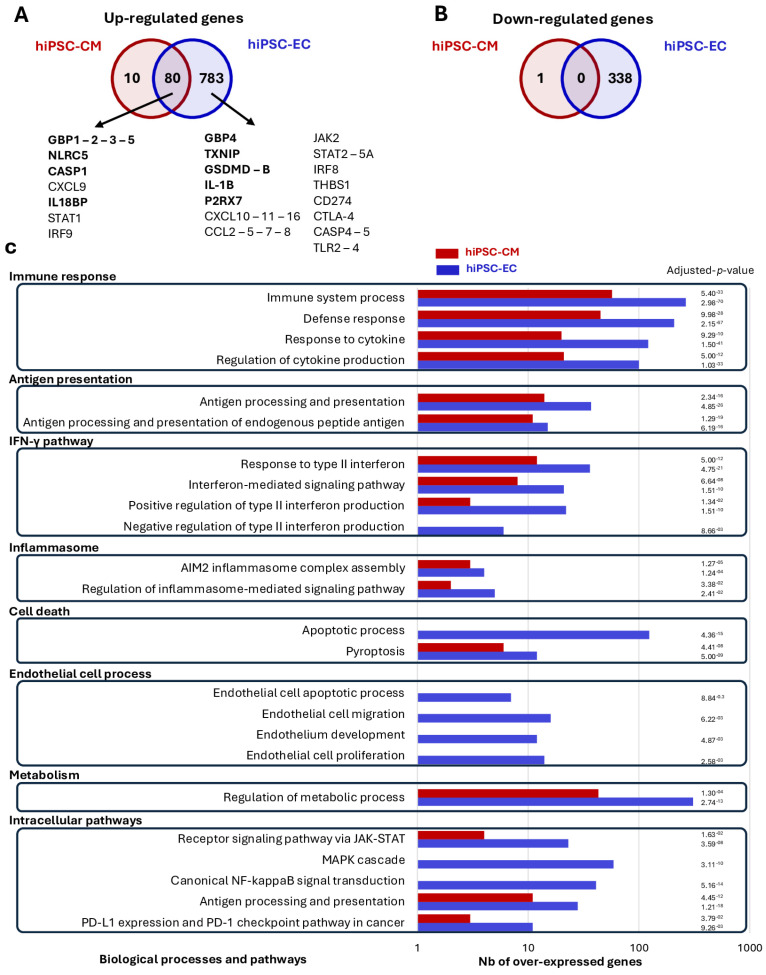
Differential Gene Expression and Pathway Enrichment in IFN-γ-treated cells. (**A**) Venn diagram illustrating upregulated genes in IFN-γ-treated hiPSC-CMs (red), hiPSC-ECs (blue), and genes common to both cell types. Genes involved in the inflammasome pathway are highlighted in bold. (**B**) Venn diagram showing downregulated genes in hiPSC-CMs (red), in hiPSC-ECs (blue), and those shared between the two cell types following IFN-γ treatment. (**C**) Functional enrichment analysis of upregulated genes in hiPSC-CMs (red bars) and in hiPSC-ECs (blue bars), categorized by associated biological processes and pathways. (bars represent the number of upregulated genes in each process, with corresponding adjusted *p*-value).

**Figure 3 cells-14-01397-f003:**
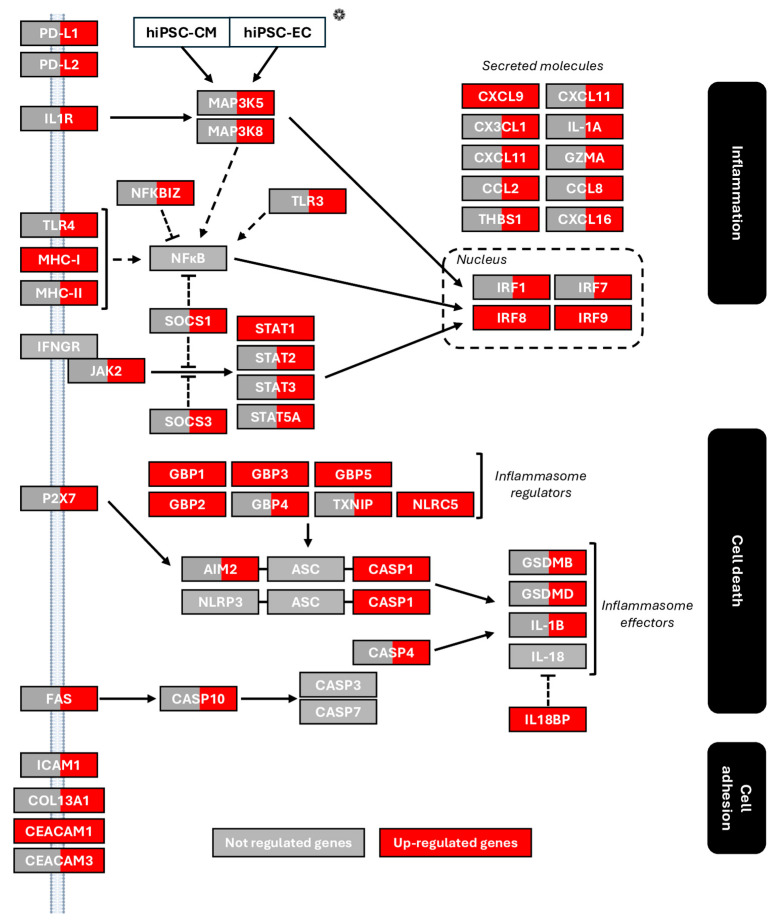
Inflammation, cell death, and adhesion pathways in IFN-γ-treated cells. The diagram illustrates the regulation of these pathways in hiPSC-CMs and in hiPSC-ECs following IFN-γ treatment, based on microarray data. ❁ For each gene, transcriptomic variation is shown in hiPSC-CMs (left side) and in hiPSC-ECs (right side) with red indicating upregulation and gray indicating no significant change. Solid arrows represent direct interactions, while dashed arrows indicate indirect interactions.

**Figure 4 cells-14-01397-f004:**
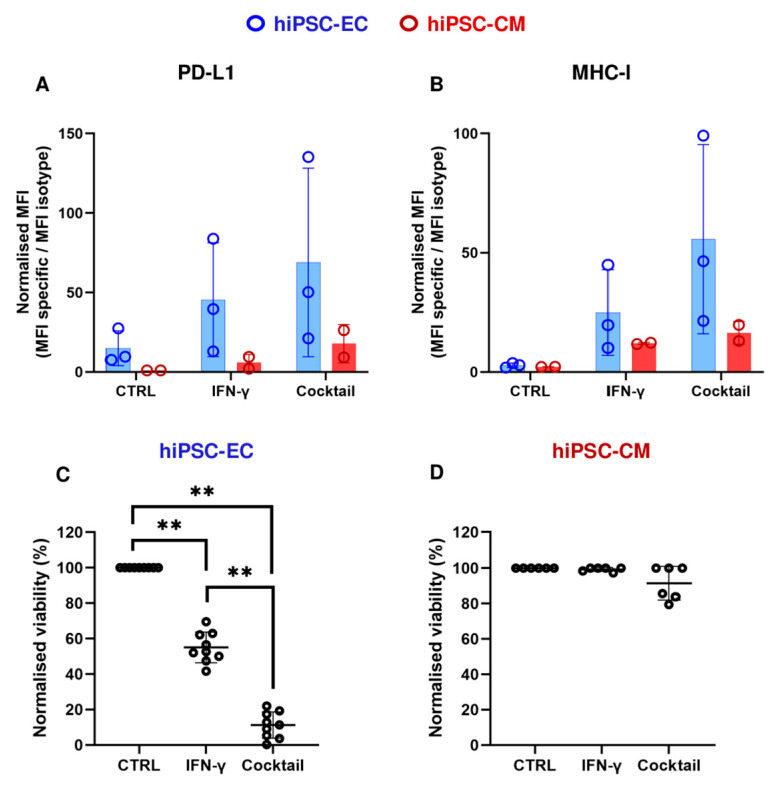
PD-L1 and MHC-I expression and cell viability in cytokine-treated cells. (**A**) Normalized mean fluorescence intensity (MFI) of PD-L1 surface expression in hiPSC-ECs (blue) and hiPSC-CMs (red) following treatment with IFN-γ, a pro-inflammatory cytokine cocktail compared to untreated controls. The cytokine cocktail included IL-1β, IL-2, IL-6, CCL5, IFN-γ, TNF-α, GZMB at the following concentrations: 10 ng/mL for all except IL-2 (2 ng/mL), TNF-α and GZMB (5 ng/mL each). (**B**) Normalized MFI of MHC-I surface expression in hiPSC-ECs (blue) and hiPSC-CMs (red) under the same conditions. (**C**) Cell viability of hiPSC-ECs and (**D**) hiPSC-CMs following treatment with IFN-γ and a pro-inflammatory cytokine cocktail. Viability values are normalized to the untreated control (CTRL). For samples following a normal distribution (as assessed by the Shapiro–Wilk test), statistical analysis was performed using *t*-test with Welch’s correction to compare cytokine-treated vs. control groups, or IFN-γ alone vs. the full cytokine cocktail. ** *p* < 0.01; Data are presented as mean ± SD (N = 2–3 independent differentiations).

**Figure 5 cells-14-01397-f005:**
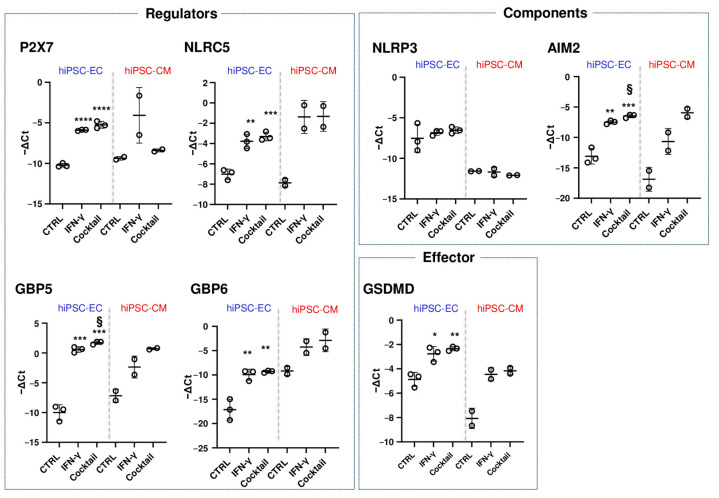
Expression of inflammasome-related genes in cytokine-treated cells. Gene expression was assessed in hiPSC-ECs (blue) and hiPSC-CMs (red) under three conditions: untreated control (CTRL), IFN-γ treatment, and cytokine cocktail treatment. The analysis included inflammasome regulators (P2X7, NLRC5, GBP5, GBP6), components (NLRP3, AIM2), and the effector GSDMD. Data are expressed as −ΔCt values, normalized to the geometric mean of housekeeping genes RPL13a and YWHAZ. For samples following a normal distribution (as assessed by the Shapiro–Wilk test), statistical comparisons were performed using *t*-test (mean ± SD, N = 2–3 independent differentiations): * *p* < 0.05; ** *p* < 0.01; *** *p* < 0.001; **** *p* < 0.0001 vs. CTRL; ^§^
*p* < 0.05 for IFN-γ vs. cytokine cocktail.

**Figure 6 cells-14-01397-f006:**
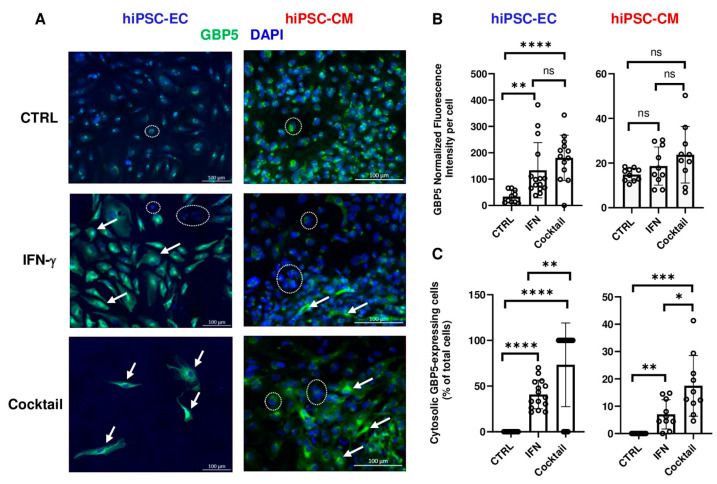
GBP5 expression in cytokine-treated cells. (**A**) Immunofluorescence staining of hiPSC-ECs and hiPSC-CMs under control (CTRL), IFN-γ, or cytokine cocktail conditions. GBP5 is shown in green; nuclei in blue (DAPI). The dotted lines indicate example of cells that do not express GBP5 in the cytosol; the arrows indicate some cells with cytoplasmic GBP5 expression. Under control conditions, GBP5 expression was restricted to perinuclear vesicles. IFN-γ treatment induced cytoplasmic GBP5 expression, predominantly in hiPSC-ECs. Exposure to the cytokine cocktail further increased the proportion of cells displaying cytoplasmic GBP5 expression. Scale bar: 100 µm. (**B**) Quantification of GBP5 expression intensity per cell for hiPSC-ECs (left) and hiPSC-CMs (right). (**C**) Quantification of cells with cytoplasmic GBP5 for hiPSC-ECs (left) and hiPSC-CMs (right). Statistical analysis was performed using *t*-test with Welch’s correction to compare cytokine-treated to control and IFN-γ alone to the cytokine cocktail. * *p* < 0.05; ** *p* < 0.01; *** *p* < 0.001; **** *p* < 0.0001; Data are presented as mean ± SD from N = 2–3 independent differentiations (n = 10 for CTRL; n = 15 for IFN-γ and cytokine cocktail in hiPSC-ECs. n = 5 for CTRL; n = 10 for IFN-γ and cytokine cocktail in hiPSC-CMs).

**Figure 7 cells-14-01397-f007:**
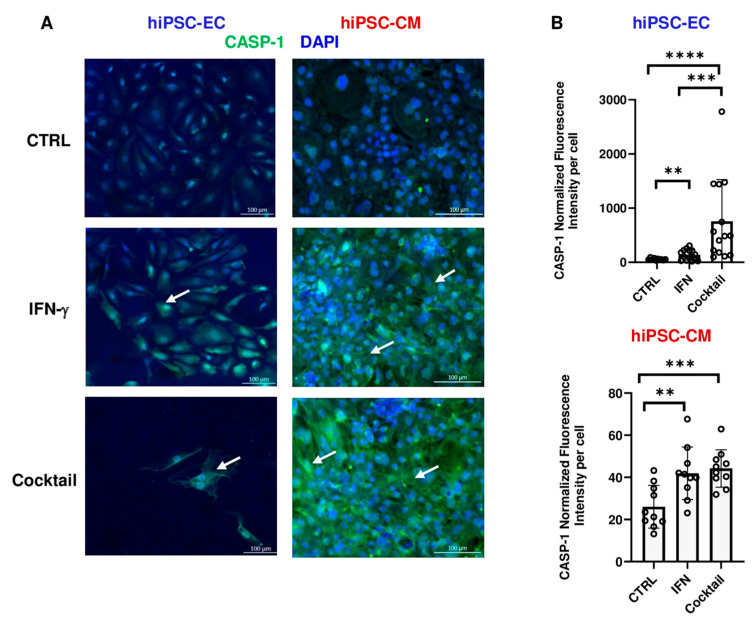
Active caspase-1 expression in cytokine-treated cells. (**A**) Immunofluorescence staining of hiPSC-ECs and hiPSC-CMs under three conditions: untreated control (CTRL), IFN-γ treatment, and cytokine cocktail treatment. Active-CASP-1 (cleaved-CASP-1) expression is visualized in green, and nuclei are stained with DAPI (blue). White arrows indicate cells with elevated cytoplasmic cleaved-CASP-1 expression. Scale bar: 100 µm. (**B**) Quantification of cleaved-CASP-1 expression, based on fluorescence intensity normalized to the total number of cells in the analyzed area, shown for hiPSC-ECs (top) and hiPSC-CMs (bottom). Data points represent technical replicates. Statistical analysis was performed using *t*-test with Welch’s correction to compare cytokine-treated to control and IFN-γ alone to the cytokine cocktail. ** *p* < 0.01; *** *p* < 0.001; **** *p* < 0.0001; Data are presented as mean ± SD from N = 2–3 independent differentiations (n = 10 for CTRL; n = 15 for IFN-γ and cytokine cocktail in hiPSC-ECs. n = 5 for CTRL; n = 10 for IFN-γ and cytokine cocktail in hiPSC-CMs).

**Figure 8 cells-14-01397-f008:**
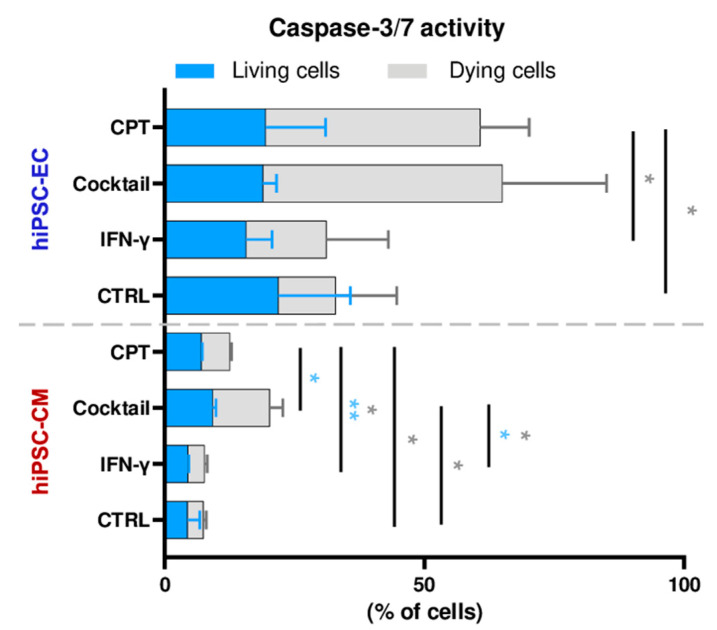
Caspase-3/7 activity in cytokine-treated cells. Percentage of caspase-3/7–active cells in living (blue) and dying (gray) populations following IFN-γ-, cytokine cocktail-, or camptothecin (CPT)-treatment. Living cells were defined by intact plasma membrane integrity; dying cells exhibited membrane compromise. CPT: positive control for cell death. Statistical analysis was performed using t tests: * *p* < 0.05, ** *p* < 0.01. Bars represent mean ± SD from 2 to 3 independent differentiations.

## Data Availability

The majority of the data generated or analyzed during this study are included in this article and its Appendix A. Additional datasets are available from the corresponding author upon reasonable request.

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
