# Peer review of "Distinct Inflammatory Responses of hiPSC-Derived Endothelial Cells and Cardiomyocytes to Cytokines Involved in Immune Checkpoint Inhibitor-Associated Myocarditis"

_cells, 2025, doi:10.3390/cells14171397_

Round 1

Reviewer 1 Report

Comments and Suggestions for Authors

Review:

Clinical studies have shown that patients with ICI myocarditis have upregulation of circulating levels of several cytokines, with varying levels in different patients.  Transcriptomic analysis shows that gene and protein expression of guanylate-binding proteins GBP5 and GBP6, which are activated by IFN gamma pathways, were significantly increased.  The objective of the study by Conte et al is to generate a human iPSC-based model of immune check point inhibitor myocarditis to examine how ICM-myocarditis associated cytokines affect the viability and inflammatory response of ECs and CMs. The investigators generate iPSCs from one healthy donor.  After performing assays to validate the iPSCs, the hiPSC-ECs and hiPSC-CMs were exposed to 10 ng/ml of IFN gamma for 72 hours.  A microarray was performed and show upregulation of genes involved in inflammation, cell death, and migration. They found that several inflammasome regulators were upregulated including GPBs. To mimic the cytokine milieu associated with check point inhibitors, the investigators exposed the hiPSC-ECs and hiPSC-CMs to IFN gamma vs a cytokine cocktail and showed death of hiPSC-ECs but not hiPSC-CMs.  Upregulation of GPBs were shown at the gene and protein level using PCR and IF, respectively. To identify mechanisms of cell death, the investigators performed caspase staining, which suggested both capsase independent and dependent pathways.

While the study presents some interesting findings, the methods could be improved. Most experiments lack sufficient sample size as detailed below. Conclusions are only made by one experimental approach. The statistical analysis also does not seem to be appropriate for the sample size. Finally, some of the figures could be enhanced.

Specific comments

  • Typical iPSC stem cell models use at least three donors. This limits the generalizability of the findings. Using a healthy donor only rather than one with myocarditis is also unclear as myocarditis does not occur in all individuals, suggesting a genetic component.
  • hiPSCs were validated by the expression of canonical gene markers but no functional assays were performed.
  • The rationale for choosing 10 ng/ml as the doses for IFN gamma as well as other cytokines are unclear and needs to be better explained. Ideally, the dose should represent the blood concentration seen in vivo. In vivo dose studies to identify doses equivalent to those observed in patients with ICI-myocarditis was not performed. No references were made to justify the use of the dose.
  • The sample sizes are small, and the statistical tests used do not seem appropriate
    1. Findings in Figure 4A (top panel) seem variable and driven by outliers. Also, only technical replicates shown with no biological replicates.
    2. Findings from Figure 5 also do not have adequate samples
    3. Figure 6: Panel C on figure 6 for the cocktail needs to be reexamined.
    4. Figure 7: The merged IF figures do not appear convincing.
  • Analysis of cell death mechanisms need to be further enhanced. The investigators rely on caspase IF staining.

Reviewer 2 Report

Comments and Suggestions for Authors
  1. Could the authors add in the abstract, as a summary, one or two sentences about the significance and impact of their own research results in the prevention of myocarditis?
  2. In the Materials and Methods section, the following sentence appears: Biologic samples and cell culture. The human induced pluripotent stem cell (hiPSC) clone ECT-06 was established by the cell reprogramming and differentiation facility (MaSC) of the Marseille Medical Genetics Institute (MMG-UMR 1251), from skin fibroblasts from a healthy male donor (aged 69). Do the authors have an ethics committee approval number or written consent from the fibroblast donor?
  3. Materials and methods, RNA extraction, lines 166-167: there is a sentence there:  RNA was treated with 166 DNase I (QIAGEN) following the manufacturer’s instructions, it is worth adding the kit number in this case so that the reader can familiarize himself with the details.
  4. Materials and methods, cell treatment, lines 158-164: Can the authors explain what influenced: a) the duration of IFN-γ (10 ng/ml, Miltenyi Biotec) administration (48 vs. 72 hours)? b) whether cells were exposed to a single proinflammatory factor (e.g., IFN, CCL5) or a combined cytokine cocktail?
  5. Materials and methods, The control group should be better defined. This section doesn't make it clear whether the controls and treated samples are separate cultures or simply replicates of the qPCR and microarray runs. In my opinion, this should be clarified to better understand the statistical significance of the entire study.
  6. Results, line 478: correct p value to p
  7. Discussion and Conclusions: in my opinion, adding data such as statistical significance and the number of replicates/trials when discussing the results would have helped the reader better understand the results of this experiment.
  8. Discussion, lines 585-594: In this section, the authors describe an interesting observation regarding NLRP3 and AIM2. They found that NLRP3 transcript levels were unchanged by IFN-γ, whereas AIM2 expression was either induced or increased by the same factor, depending on the cell type. Is it possible for the authors to add information about how IFN-γ concentration affected the rate and dynamics of change in the AIM2 transcriptome?
Comments on the Quality of English Language

The entire manuscript should be proofread by a native speaker to generally correct style and errors resulting from non-English speaking authors.

Reviewer 3 Report

Comments and Suggestions for Authors

This manuscript by Conte et al. describes the use of cardiomyocyte and endothelial cell models derived from the same iPSC line to study the effect of inflammatory cytokines implicated in checkpoint inhibitor-induced myocarditis. The main finding shown is that endothelial cells are more sensitive to the effect of IFN-γ and inflammatory cytokines than cardiomyocytes, which provides relevant information useful for developing diagnostic and therapeutic tools.

The article is well-written, presents the results coherently, and draws appropriate conclusions. However, I have some questions and comments for the authors in order to improve the manuscript and provide potential readers with the necessary information for a better understanding.

I would like to rise some questions regarding Figures 6 and 7 of the manuscript:

1) although the images shown are representative of the experiments performed, I find the low number of hiPSC-EC cells in the Cocktail condition surprising. This may be due to low cell viability (shown in Figure 4C), but the proportion of cells with respect to the other conditions seems lower than expected

2) Figure 6C shows the percentage of cells with increased cytosolic expression of GBP5 (note that the title in the vertical axis is written GPB5). How was positivity for expression determined? Looking at the images, practically all cells show fluorescence in the cytoplasm at very similar levels, and the cells marked with arrows as cells with increased cytosolic GBP5 do not show a much higher fluorescence intensity than others in the vicinity.

In the Apendix, data are shown on the effect of cytokine treatment on viability of hiPSC-EC, hiPSC-CM and AC16 (Figure A1), as well as on the protein expression level of MHC-I and PD-L1 (Figure A2). Did the authors perform this analysis for HCMEC cells? Additionally, in the Figure A3 there are only data corresponding to CTRL, IFN-γ, and cocktail treatments, although in the caption it is mentioned also camptothecin treatment.

The Appendix shows data on the effect of cytokine treatment on the viability of hiPSC-ECs, hiPSC-CMs, and AC16 cells (Figure A1), as well as on the protein expression levels of MHC-I and PD-L1 (Figure A2). Did the authors perform these analyses for HCMEC cells? Furthermore, Figure A3 only includes data for CTRL, IFN-γ, and cocktail treatments, although camptothecin treatment is also mentioned in the caption.

Finally, a repeated typo can be found throughout the manuscript: IFN-g instead of IFN-γ in caption of Figure 2, Figure 3, Figure 4, Figure 8, Table S1, Figure A3, Figure A4.
